# Experimental and Simulation Study on the Effect of Reduction Pretreatment on the Void Healing of Heavy Plate

Zhen Ning [1], Xue Li [1], Hongqiang Liu [1,2] , Qingwu Cai [1] and Wei Yu [1,*]

1. Collaborative Innovation Centre of Steel Technology, University of Science and Technology Beijing, Beijing 100083, China; zhenning410@163.com (Z.N.); 18345065610@163.com (X.L.); liuhongqiang@hbisco.com (H.L.); caiqw@nercar.ustb.edu.cn (Q.C.)
2. HBIS Group Co., Ltd., Shijiazhuang 050000, China
* Correspondence: yuwei9451@126.com or yuwei@nercar.ustb.edu.cn; Tel.: +86-13001183873

**Abstract:** In addition to eliminating voids in a billet, the reduction pretreatment (RP) process can refine the austenite structure after reheating and can deform the residual voids, which affect the void healing behavior during insulation before rolling, and then can affect the final quality of the steel plate. In this study, the effect of the RP process on the evolution of voids in billets and steel plates was investigated by experimental and numerical simulation. RP and non-RP billets were reheated and rolled into 60 mm-thick steel plates, respectively, and the evolutionary behavior of voids within the plates was analyzed using ultrasonic flaw detection methods. Finite element method (FEM) and phase-field method (PFM) were used to investigate the mechanism behind improving the quality of steel plates under the RP process. The simulation results show that, in addition to directly closing the voids through plastic deformation, the RP process can promote the diffusive healing behavior during insulation before rolling by refining the reheated austenite organization and by increasing the curvature of the void surface. Due to the reduction in voids inside the steel plate, the impact energy of the RP steel plate is higher.

**Keywords:** reduction pretreatment; void healing; phase-field method; ultrasonic testing

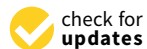



## 1. Introduction

Due to volume shrinkage that occurs during the solidification process, voids in a billet are inevitable. These voids will adversely affect the final quality of the workpiece, and it is necessary to reduce or eliminate them. The elimination of these voids has aroused the widespread interest of researchers.

Hot deformation (hot rolling and hot forging) is usually employed to reduce voids in the workpiece [1–4]. Appropriate thermal processing process parameters are critical to eliminating voids. Therefore, researchers have carried out many studies on the influence of process parameters on void closure [5–8]. Previous research shows that, in thermal deformation, it is necessary to achieve a sufficient pass reduction to eliminate voids [4]. When the size of the workpiece is large, due to the problem that the workpiece is difficult to bite, this pass reduction is difficult to achieve. Compared with isothermal deformation, when there is a temperature gradient between the workpiece's surface and center, it is easier to eliminate the voids in the workpiece center [5,9,10]. The temperature gradient causes a difference in the deformation resistance between the surface and the core of the workpiece and leads to greater plastic deformation in the billet center. Therefore, if it is difficult to further increase the pass reduction, increasing the plastic deformation of the workpiece center through the temperature gradient to eliminate the voids becomes a good choice.

During billet solidification, the surface and core of the billet produce a large temperature gradient. To use this temperature gradient to improve the quality of the billet, researchers have developed various billet reduction processes. For example, the porosity

control of casting slab (PCCS) process developed by Sumitomo Metal Industries [11], the Heavy Reduction Process to Improve Segregation and Porosity (HRPISP) process developed by Zhao et al. [12], the two-stage continuous heavy reduction technology developed by Ji et al. [13], and the reduction pretreatment (RP) process developed by Li et al. [14]. The common feature between these technologies is the use of the temperature gradient generated by solidification waste heat to produce a relatively large equivalent strain in the billet center, to eliminate void-type defects in the billet, to break the coarse as-cast structure, and to have a positive effect on improving the quality of the billet. For these processes, researchers have carried out much follow-up research. For example, through experiments and simulations, Cheng et al. [15] studied the influence of rolling passes on void closure. Their research shows that the slab quality improvement effect of the single-pass reduction mode is better than that of the multi-pass reduction mode. Gong et al. [16] studied the effect of the hot-core heavy reduction process on precipitates and austenite structure of billets during reheating. It was found that randomly distributed nano-precipitates were generated in the billet, and the reheated austenite structure was significantly refined. In order to study the evolution of austenite during heavy reduction, Yang et al. [17] established a dynamic recrystallization kinetic model in the range of process parameters with a temperature of 1173–1573 K and strain rate of 0.001–0.1 s$^{-1}$, and applied this model to predict the flow–stress curve. Li et al. [18] used the finite element method to study the effect of the heavy reduction process on the surface cracking of the billet. It was found that the risk of billet surface cracking increased with the increase in reduction and that the critical reduction was 59%.

The ultimate goal of improving billet quality is to improve product quality. At present, there exist some studies on the influence of the RP process on subsequent thermal deformation processes, but they all focus on the microstructure and properties of products [19]. Research on the influence of the RP process on void evolution behavior in the subsequent hot working process remains absent. The effect of the hot rolling process on the elimination of voids is divided into two main parts. One is the rolling process, which through plastic deformation directly eliminates voids; the other is diffusive healing during the insulation before rolling. The RP process can change the shape of voids and the reheated microstructure in the billet. According to previous studies, void diffusive healing is controlled by the vacancy concentration difference between the void and the matrix [20], and the void shape [21] and grain boundary density [22] affect the diffusion behavior of the vacancy on the void surface and the matrix. Therefore, this paper studies the influence of the RP process on void diffusive healing during the insulation before rolling. As a mesoscopic simulation method, the phase-field method can simulate the microstructure evolution process. At present, it has been applied in void evolution [23] and grain growth [24]. Since the high-temperature void diffusive process is difficult to observe directly through experiments, the phase-field method is used to qualitatively study the effect of the RP process on void diffusive healing.

The main content of this article is to further study the influence of the RP process on the thick plate quality based on the previous research [14,25]. For this purpose, two billets were produced in the laboratory, and the RP process was applied to one of them. After that, the two billets were rolled into thick plates. The influence of the RP process on the void closure of the billet and the thick plate was studied through macrostructure observation and ultrasonic testing (developed by Li et al. [26]). Finite element and phase-field simulation were used to study the effect of the RP process on eliminating voids. Moreover, the mechanical properties of the steel plate were measured.

## 2. Materials and Methods

### 2.1. RP Process

The steel selected for this research is 42CrMo steel, which has the composition (wt pct) 0.384 pct C-0.737 pct Mn-0.236 pct Si-1.13 pct Cr-0.186 pct Mo-0.0065 pct P-0.0041 pct S. The billets were cast through a vacuum induction furnace (Northeastern University, Shengyang,

China) at a smelting temperature of 1833 K. The billet size was 150 mm × 150 mm × 300 mm. After stripping, Billet A was cooled in the air to room temperature. Billet B was rolled using a rolling mill (Northeastern University, Shengyang, China) with a 750-mm roller diameter, 0.35 m/s roll speed, and a 126 mm roll gap after the surface temperature cooled to 1223 K. The billet was cut into a casting part and rolling part from the middle, as shown in Figure 1. To observe the macrostructures, a cross-sectional sample with a thickness of 15 mm was cut from the casting part. After taking pictures of the macrostructure with the camera, the ultrasonic testing sample was cut from the cross-sectional sample. The cutting position is illustrated in Figure 1. The sample size was 100 mm × 70 mm × 15 mm. After the upper and lower surfaces of the sample were finely ground, ultrasonic testing was carried out on the PVA Sam 300 ultrasonic microscope (PVA TePla, Westhausen, Germany).

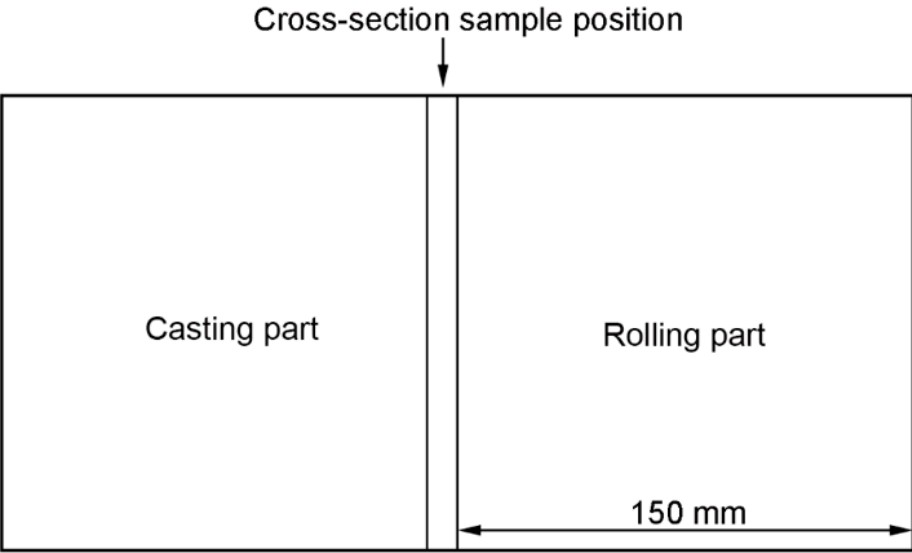

**Figure 1.** Position of macrostructure sample. Casting part: observe the voids in the billet; rolling part: rolled into a steel plate.

### 2.2. Rolling Process

After holding at 1523 K for 1 h, the rolling parts of Billets A (non-RP billet) and B (RP billet) were rolled into a 60 mm thick plate—the rolling schedule is shown in Table 1. After rolling, the plates were air-cooled to room temperature. The U-shaped notch toughness specimens were machined at one-half and one-quarter thickness of the plates. The sampling direction is the cross direction. The toughness was measured according to standard GB/T 229-2007. The microstructure sample was cut near the notch of toughness specimens. After observing the microstructure with optical microscopy (ZEISS, Gottingen, Germany), ultrasonic testing was completed on the microstructure sample.

**Table 1.** Rolling schedule.

|  | Pass No. | 1 | 2 | 3 | 4 | 5 | 6 |
|---|---|---|---|---|---|---|---|
|  | Temperature, K | 1373 | 1353 | 1303 | 1273 | 1253 | 1223 |
| **Plate A (non-RP billet)** | $t_0$, mm | 150 | 126 | 110 | 95 | 82 | 70 |
|  | $t_f$, mm | 126 | 110 | 95 | 82 | 70 | 60 |
| **Plate B (RP billet)** | $t_0$, mm |  | 126 | 110 | 95 | 82 | 70 |
|  | $t_f$, mm |  | 110 | 95 | 82 | 70 | 60 |

### 2.3. Finite Element Method (FEM) Simulation for the RP Process

To explain the effect of the RP process on void closure in the billet, a 3D thermo-mechanical coupling model was developed using Abaqus software. Since the difference between two plates lies only in the first pass rolling and the RP process, the model was only used to simulate these two processes. The material parameters of experimental steel were shown in Table 2. The empirical model established by Chadha et al. [27] was used to describe the thermal-plastic behavior of 42CrMo steel:

$$\sigma = 2136.313 \times e^{-0.00243T} \times \varepsilon^{0.2315} \times \dot{\varepsilon}^{0.1215} \times e^{\frac{0.0001}{\varepsilon}} \times (1 + \varepsilon)^{-0.001T} \times e^{0.3235\varepsilon} \quad (1)$$

where $\sigma$ is the stress [MPa], $\varepsilon$ is the strain, and $\dot{\varepsilon}$ is the strain rate.

**Table 2.** Material parameters of experimental steel.

| | |
|---|---|
| Density/(kg m$^{-3}$) | 7800 |
| Heat convection coefficient/(W m$^{-2}$ K$^{-1}$) | 50 |
| Specific heat/(kJ kg$^{-1}$ K$^{-1}$) | 0.72 |
| Elastic module/GPa | $-0.1015 \cdot T + 234.15$ |
| Poisson's ratio | $0.00006 \cdot T + 0.2756$ |
| Element type | C3D8 |
| Element size/mm | 5 |

*T*, Temperature.

The simulated temperature of the first pass rolling was 1323 K. The temperature distribution of the RP process was calculated by using a 3D FEM model [28], which is shown in Figure 2.

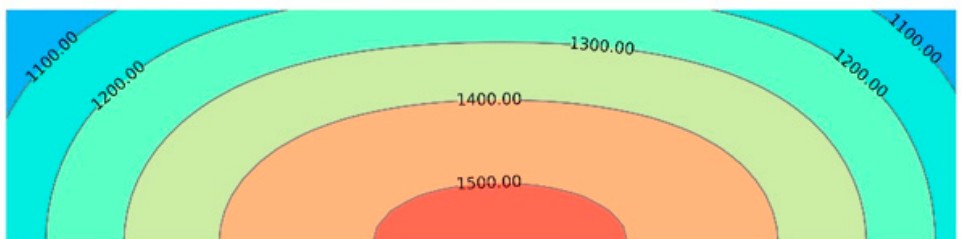

**Figure 2.** Temperature condition of RP process.

The reduction pretreatment process can refine the microstructure of the billet. Due to the microstructure inheritance, the grain size of the RP billet after reheating will be much smaller than the original slab, and the grain boundary density of the matrix will be greater [19]. At the same time, the RP process deformed the voids in the billet, and these two parts will affect the diffusive healing behavior of the void. To research the RP process' effect on void diffusive healing, the phase-field method (PFM) was used to qualitatively simulate the void diffusive healing process under different grain sizes and void shapes. The void healing process can be considered a diffusion-controlled phase change process. The healing rate is related to the solubility of the vacancy and the diffusion speed in the matrix. Here the void healing process was simulated using the model developed by Millett et al. [29]. The free energy function of the system consisting of both a matrix and a void phase is described in terms of the vacancy concentration field and the order parameter field $\eta$ as follows:

$$F = \int \left[ h(\eta) f_v^m(c_v) + g(\eta) + \frac{\kappa_c^2}{2} |\nabla c_v|^2 + \frac{\kappa_\eta^2}{2} |\nabla \eta|^2 \right] d^3 x \quad (2)$$

where $h(\eta)$ is the shape function, can be expressed as follows:

$$h(\eta) = (\eta - 1)^2 (\eta + 1)^2 \tag{3}$$

$f_v^m(c_v)$ is the free energy contribution of vacancy concentration and is written as follows:

$$f_v^m(c_v) = c_v \Delta E_v^f + k_B T [c_v \ln(c_v) + (1 - c_v) \ln(1 - c_v)] \tag{4}$$

$g(\eta)$ is the potential term:

$$g(c_v, \eta) = B_1 (c_v - 1)^2 \eta^2 + B_2 \left(c_v - c_v^{\text{eq}}\right)^2 (\eta - 1)^2 \tag{5}$$

where $B_1$ and $B_2$ are constants, and $c_v^{\text{eq}}$ is the thermal equilibrium vacancy concentration, which can be written as follows:

$$c_v^{\text{eq}} = \exp\left(-E_v^f / k_B T\right) \tag{6}$$

$\kappa_c$ and $\kappa_\eta$ are the gradient energy coefficients of vacancy concentration and the phase field.

The evolution equations of the order parameter field and concentration field can be written as follows:

$$\frac{\partial \eta}{\partial t} = -L_\eta \frac{\delta F}{\delta \eta} \tag{7}$$

$$\frac{\partial c_v}{\partial t} = \nabla \cdot \left(M_v \nabla \frac{\delta F}{\delta c_v}\right) - S_v^{\text{GB}} (1 - \phi^2) \left(c_v - c_v^{\text{eq}}\right) \tag{8}$$

where $L_\eta$ and $M_v$ are the mobility of interface and vacancy. $S_v^{\text{GB}}$ is a constant and illustrates the effect of grain boundaries on the healing of voids. $\phi^2$ describes the location of the grain boundaries, and the evolution of $\phi$ is calculated by the grain growth model developed by Fan and Chen [24].

$$\frac{\partial \phi_i}{\partial t} = -L\left(-\phi_i + (\phi_i)^3 + 2\phi_i \sum_{i \neq j}^{N} (\phi_i)^2 - \kappa_\phi \nabla^2 \phi_i\right), i = 1, 2, \ldots, N \tag{9}$$

where $L$ is the phase-field mobility and $\kappa_\phi$ is the gradient energy coefficient.

The initial organization of the simulation is shown in the Figure 3. The radius of the original void is 10 $\Delta$x, the size of the representative volume element (RVE) is 256 $\Delta$x $\times$ 256 $\Delta$x, and $\Delta$x = 1 is the mesh size of finite element difference method for solving the phase-field model.

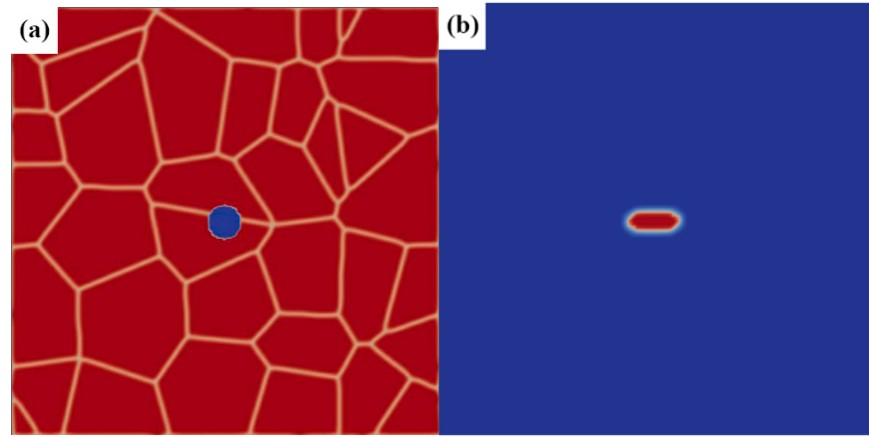

**Figure 3.** Initial organization of the phase-field method (PFM) simulation: (**a**) the grain-size effect; (**b**) the void-shape effect.

Due to the lack of actual material parameters, the phase-field simulation of this study uses dimensionless parameters to qualitatively simulate the void diffusive healing. The following parameter values were used: $B_1 = 1.0$, $B_2 = 1.0$, $L_\eta = 1.0$, $M_v = 1.0$, $\kappa_c = 1.0$, $\kappa_\eta = 1.0$, $\kappa_\phi = 2.0$, and $L = 1.0$.

## 3. Results and Discussion

The defect characterization of billets is shown in Figure 4. It can be seen from Figure 4a,c that, compared with Billet A, there are no macro-voids in the center of Billet B. In this study, the "macro-voids" and the "micro-voids", respectively, refer to the voids that can or cannot be observed by the naked eye directly. The pictures of the micro-void distribution show that there are fewer micro-voids in Billet B than in Billet A, and most of them are concentrated in the billet center. The results indicate that the RP process can improve the quality of the billet and provide better raw material for steel plate production.

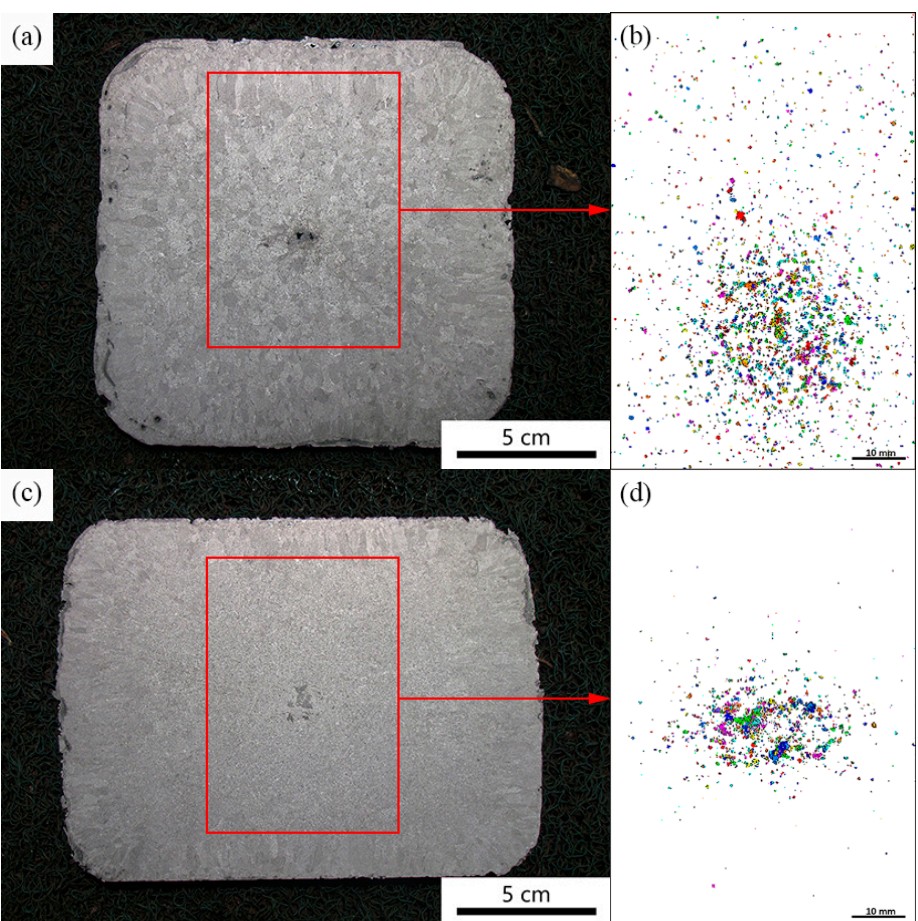

**Figure 4.** Defect characterization of billets: (**a**) macro-void distribution of Billet A; (**b**) micro-void distribution of Billet A; (**c**) macro-void distribution of Billet B; (**d**) micro-void distribution of Billet B.

To quantitatively study the influence of the RP process on the billet quality, the Image-Pro Plus software was used to measure the total number and average size of the micro-voids in the billet, and the size distribution of micro-voids was counted, as shown in Figure 5. The total number of micro-voids in Billet A and Billet B are 1746 and 622, and their average sizes are 373 μm and 390 μm, respectively. The measurement results show that the RP process significantly reduces the number of micro-voids in the billet while increasing the average size. This situation can be explained by the size distribution of micro-voids. As shown in Figure 5, the RP process significantly reduces the number of micro-voids below 500 μm in the billet, so that the proportion of small-sized micro-voids in the total number

of micro-voids is reduced considerably, increasing the average void size. In the meantime, the RP process causes some large-size voids to appear in Billet B, for two reasons: (1) the partially compressed macro-voids formed the micro-voids with large-size; (2) aggregation of small-size micro-voids formed the large-size micro-voids.

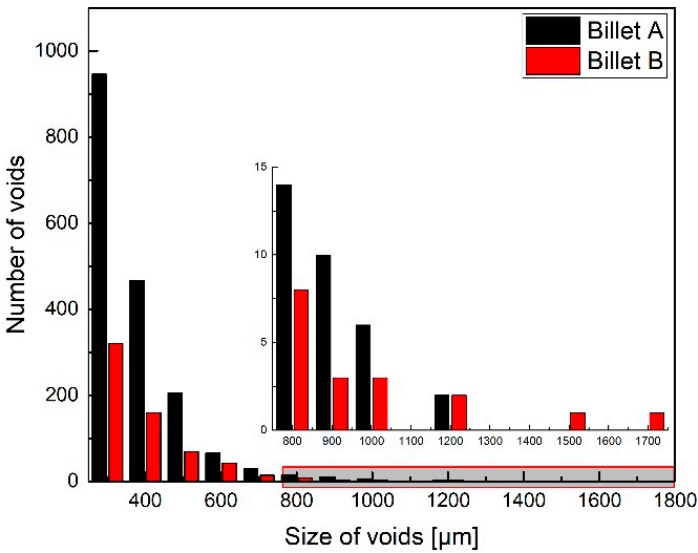

**Figure 5.** Size distribution of micro-voids in billets.

The simulation results of the first pass rolling of Billet A and the RP process of Billet B are shown in Figure 6. To explain the influence of the RP process on the void closure effect, based on the results from Abaqus software, the hydrostatic integration ($Q$) [30] was calculated using the following formula:

$$Q = \int_0^{\varepsilon_{eq}} -T_X d\varepsilon \tag{10}$$

where $Q$ is the hydrostatic integration; $\varepsilon_{eq}$ is the equivalent strain; and $T_X$ is the stress triaxiality ratio, as follows:

$$T_X = \frac{\sigma_h}{\sigma_{eq}} \tag{11}$$

where $\sigma_h$ is hydrostatic pressure and $\sigma_{eq}$ is equivalent stress. A larger $Q$ value indicates better porosity closure during the hot deformation process. The results of $Q$ are shown in Figure 6b,d.

It can be seen from Figure 6, compared with the first pass of Plate A, that the RP process can enhance the equivalent strain and $Q$ of the billet center, which means a better void closure effect. This is because, during the RP process, the temperature gradually rises from the billet surface to the billet center, which leads to a gradual decrease in deformation resistance. During the RP process, the metal with high deformation resistance near the billet surface can transfer more force to the center, resulting in a higher strain and void closure effect in the billet center.

The ultrasonic testing results for the plates are shown in Figure 7. Comparing Figures 5 and 7, it can be seen that the number and size of the voids in the steel plate are significantly reduced compared with the billet. Isothermal rolling can eliminate the voids in the billet and can improve the quality of the plate. In the meantime, from Figure 7a,b it can be seen that there are some voids with a large size and a near-circular shape in Plate A and only some small voids in Plate B. This indicates that, in the case of the same total deformation, the RP process can effectively reduce the number of voids in the steel plate and can improve the quality of the steel plate.

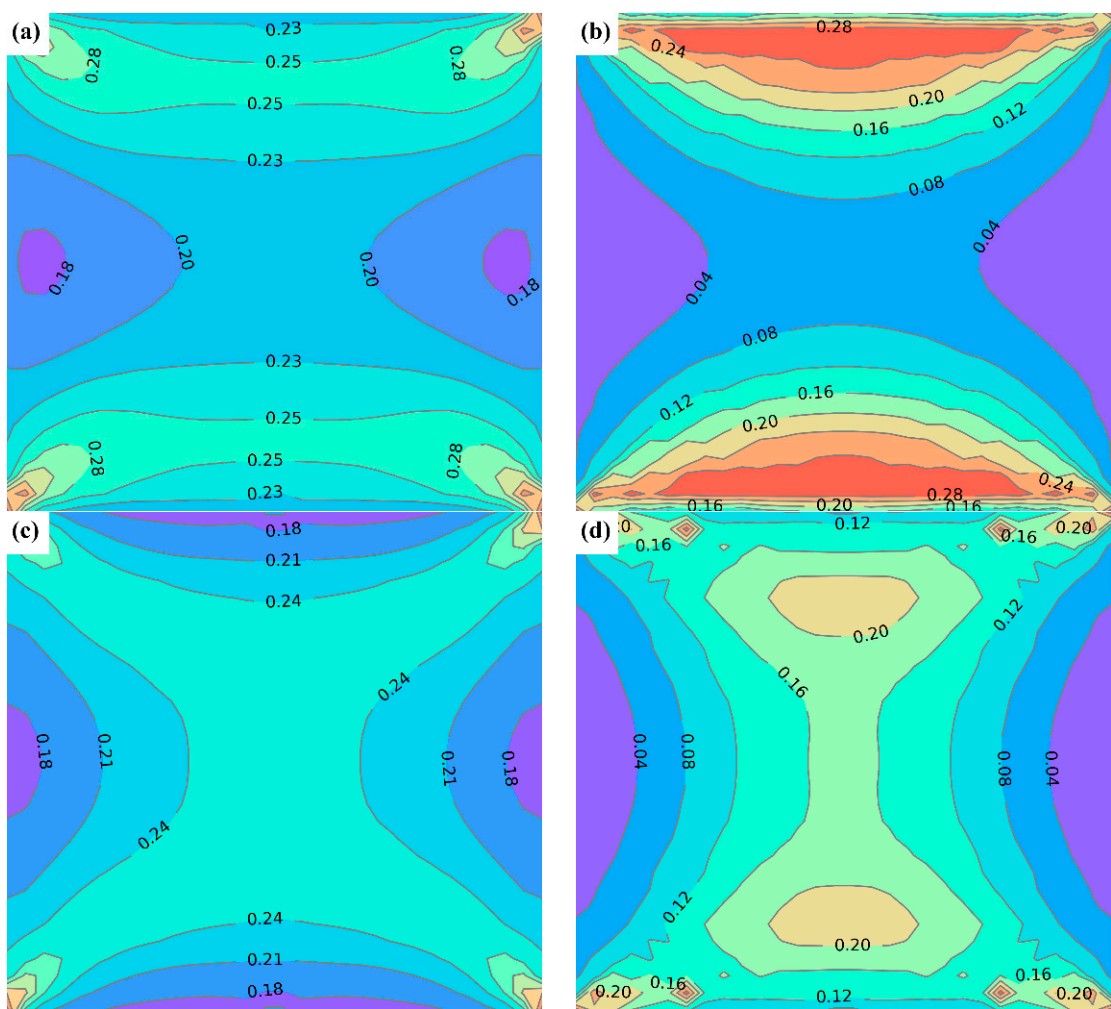

**Figure 6.** Simulation results of first pass of Plate A and RP process of Plate B: (**a**) equivalent strain of first pass of Plate A; (**b**) *Q* of first pass of Plate A; (**c**) equivalent strain of RP process of Plate B; (**d**) *Q* of RP process Plate B.

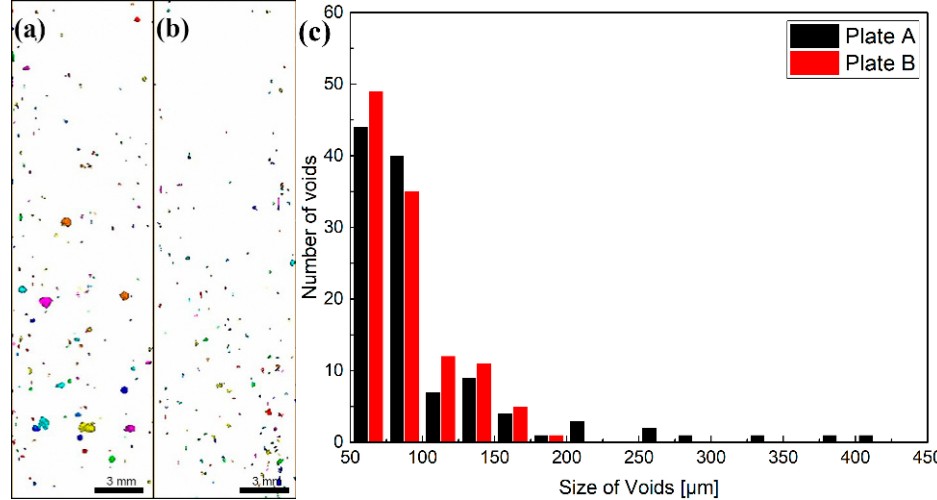

**Figure 7.** Ultrasonic testing results of plates: (**a**) Plate A; (**b**) Plate B; (**c**) size distribution of voids in plates.

The size distribution of voids in the plates is illustrated in Figure 7c. Comparison of the size distribution between voids in the plates and billets can show that there are large-size voids in both Billets A and B, and after six passes of isothermal rolling, the large-size voids in Billet A are not eliminated. However, the large-size voids in Billet B are eliminated by five passes of isothermal rolling, which shows that the RP process not only improves the quality of the billet but also affects the subsequent steel plate production process.

To eliminate the voids in the material, in addition to direct closing of the voids by plastic deformation, the voids can be healed by high-temperature diffusion. The elimination of voids during the hot rolling process can be divided into two parts. The rolling process is the void plastic closure part, and the insulation process before rolling is the void diffusive healing part. In order to study the effect of the RP process on void healing, the phase-field method is used to qualitatively simulate the void high-temperature diffusion healing process. The simulation results are as follows. The void diffusive healing process is shown in the Figure 8. It can be seen that the void size decreases with the simulation time until it disappears. According to the phase-field simulation results, the changes in the void radius with simulation time under different grain numbers in the simulation domain are counted, as shown in the Figure 9. It can be seen from Figure 9 that, when the effect of the grain boundary is not considered, the void healing rate is much slower than when the grain boundary effect is considered. In the meantime, the void healing rate is accelerated with the number of grains in the simulation domain. This is because part of the driving force of the void healing process comes from the vacancy concentration difference between the matrix and the void: the greater the vacancy concentration difference between the matrix and the cavity, the faster the cavity healing speed. When the grain boundary is not considered, the decrease in vacancy concentration in the matrix can only depend on vacancy diffusion. The grain boundary can absorb the vacancies in the matrix, can reduce the concentration of vacancies in the matrix, and can increase the driving force of void healing. Therefore, the existence of a grain boundary can accelerate void healing. Moreover, the higher the grain boundary density, the more vacancy absorption sources in the matrix, the faster the vacancy concentration in the matrix decreases, and the greater the driving force of void healing.

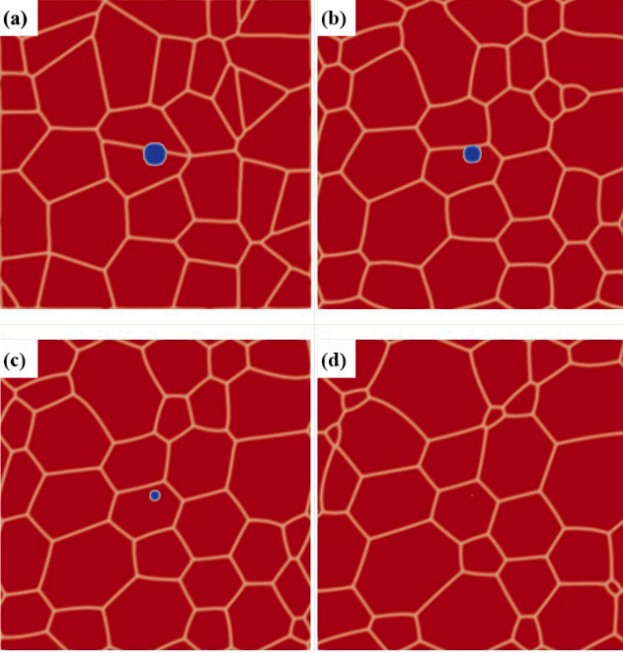

**Figure 8.** Evolution of void morphology with simulation time (36 grains): (**a**) time = 0; (**b**) time = 25; (**c**) time = 50; (**d**) time = 75. Since the phase-field method uses dimensionless parameters, the simulation results in Figure 8 are also dimensionless.

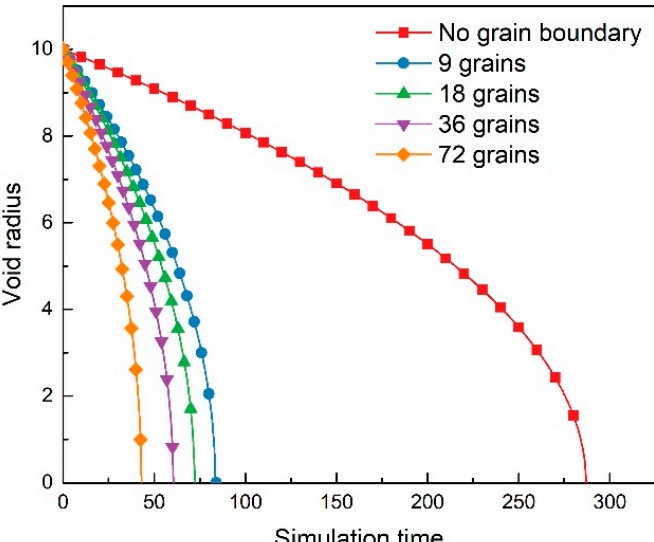

**Figure 9.** Effect of the number of grains in the simulation domain on the evolution of voids. Since the phase-field method uses dimensionless parameters, the simulation results in Figure 9 are also dimensionless.

The evolution of the deformed void with simulation times is shown in Figure 10. As seen from the Figure 10, the deformed void starts to shrink inward from the long axis until the length of the long and short axes is the same, when it starts to decrease evenly. This is because the deformed void surface has a curvature difference, and the driving force generated by the curvature difference makes the long axis healing rate faster. The effect of the void long-to-short axial ratio on the void healing rate is shown in the Figure 11. It can be seen from Figure 11 that the healing rate of the void with a larger long-to-short axial ratio is faster, indicating that the void with larger deformation is healing faster.

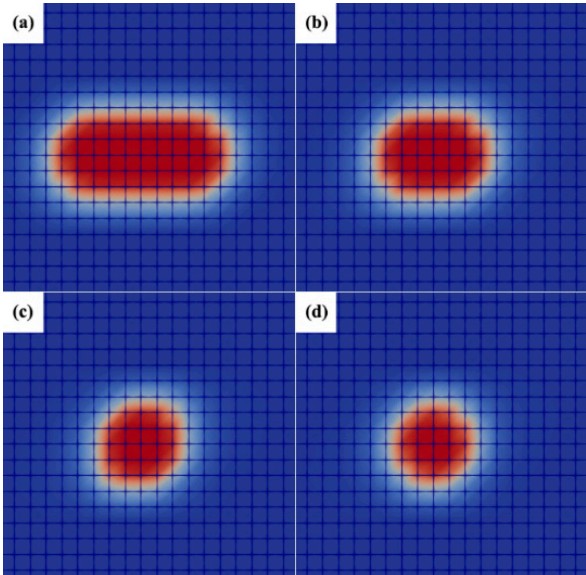

**Figure 10.** The healing process of the deformed void (8:2), (**a**) time = 0; (**b**) time = 100; (**c**) time = 150; (**d**) time = 175. Since the phase-field method uses dimensionless parameters, the simulation results in Figure 10 are also dimensionless.

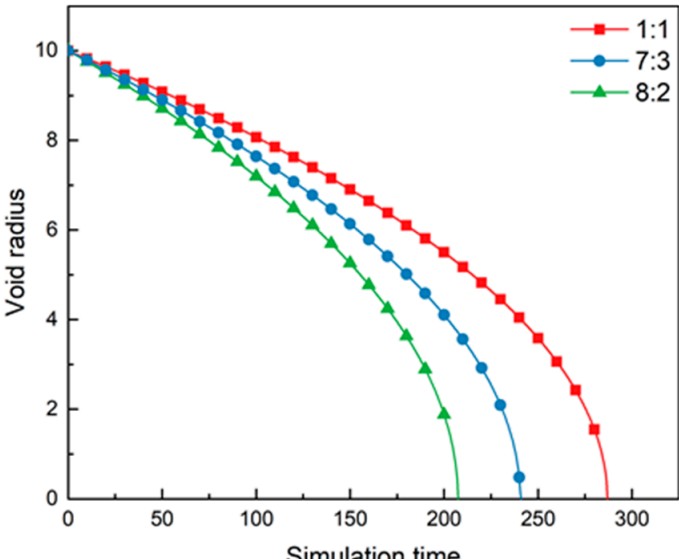

**Figure 11.** Effect of the void long and short axis ratio on the void healing rate. Since the phase-field method uses dimensionless parameters, the simulation results in Figure 11 are also dimensionless.

To summarize, the RP process mainly plays the following roles in eliminating voids in a hot-rolled steel plate. Firstly, due to the temperature gradient between the billet surface and the core, the RP process increases the deformation of the billet core and greatly reduces the cavity in the billet. Secondly, due to the microstructure inheritance, the austenite grain of the RP billet after reheating is finer than that of the non-deformed billet. The grain boundary provides many vacancy absorption sources for the void high-temperature diffusive healing, increasing the free energy difference between the matrix and voids and promoting the voids' diffusive healing. Finally, the RP process changes the void shape, resulting in a significant curvature difference on the void surface. This curvature difference drives the material to transfer from the low curvature to the position of high curvature to speed up void healing. It is worth noting that, with the progress of grain growth and hole healing, the grain refinement caused by microstructure inheritance and the interface curvature difference obtained by deformation will disappear, and the effect of the above two parts will vanish.

As shown in Figure 6, the RP process causes large plastic deformation in the billet center. Although some large-size voids are not fully closed, plastic deformation refines the matrix structure and produces a significant curvature difference on the void surface. These two parts greatly improve the void diffusive healing rate during the insulation before rolling, which explains the phenomenon that large voids in steel B are eliminated when the total deformation is the same.

The microstructures at the quarter and half thicknesses of the plate are given in Figure 12. Both Plates A and B show typical bainite and martensite structures. In Figure 13, the impact energies at the quarter and half thicknesses in Plate B are increased by 13.6 and 11.4 J, respectively, compared with the values in Plate A. Since the microstructures of the two experimental steels are the same, it can be determined that the difference in the mechanical properties of the plates do not come from the microstructure but the difference in the voids. As mentioned in the preceding part, the plate with the RP process is characterized by fewer voids, which results in relatively good toughness. The RP process has proven to be an effective means of improving the mechanical properties of steel plates.

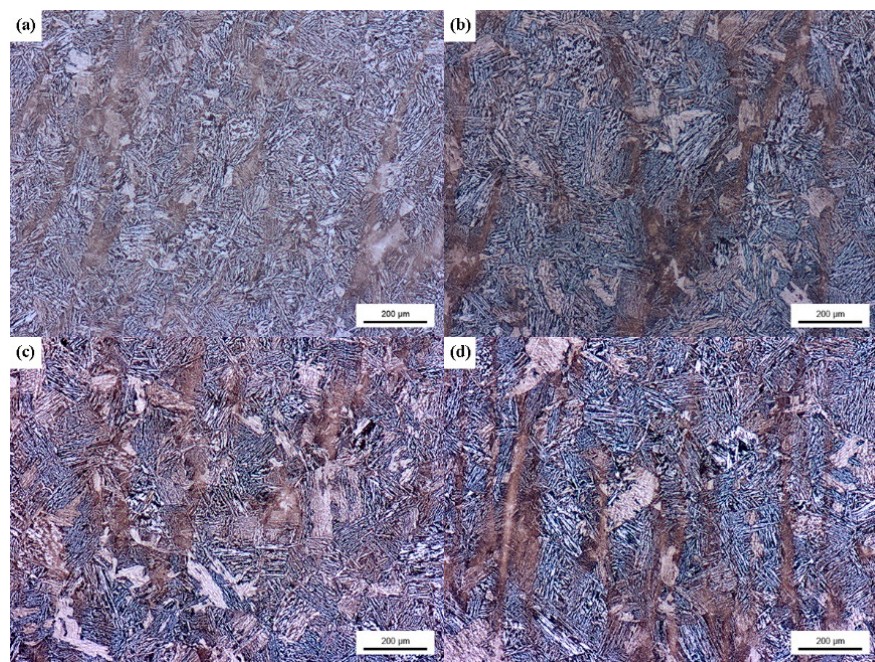

**Figure 12.** Microstructure of plates: (**a**) 1/4 thickness of Plate A; (**b**) 1/2 thickness of Plate A; (**c**) 1/4 thickness of Plate B; (**d**) 1/2 thickness of Plate B.

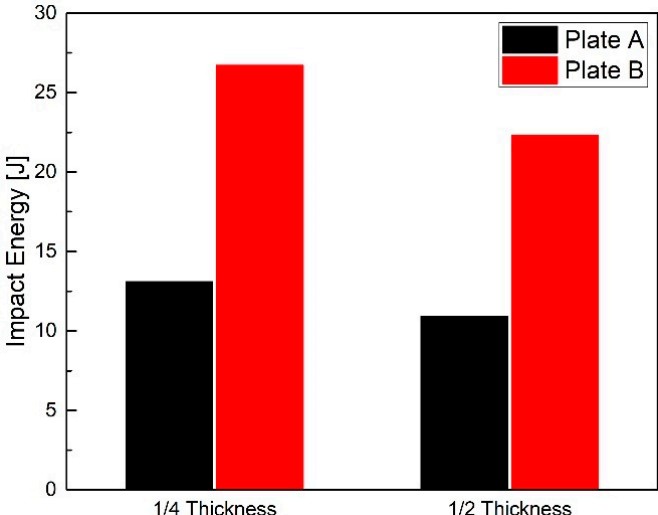

**Figure 13.** Impact energy of Plate A and Plate B.

## 4. Conclusions

In this work, to investigate the influence of the RP process on the quality of thick plate, two billets with/without the RP process were rolled into steel plates. Through numerical simulation and experiments, the influence of the RP process on eliminating internal cavities in steel was studied, and the following main conclusions were obtained:

1.  The RP process can enhance the quality of a billet: macro-voids are eliminated, and the amount of micro-void is significantly reduced.
2.  Grain size and void shape affect the void diffusive healing rate. The smaller the grain size of the matrix, the greater the curvature difference of the cavity surface and the faster the void diffusive healing rate.
3.  The RP process can promote void healing during insulation before rolling. The RP process refines the microstructure of the reheated billet, increases the curvature

difference of the void surface, and greatly promotes void diffusive healing during insulation before rolling.

4. The RP process significantly improves the quality of the plate, reduces the internal voids of the plate, and enhances the toughness.

**Author Contributions:** Conceptualization, W.Y. and Q.C.; methodology, Z.N. and X.L.; software, H.L.; validation, Z.N.; investigation, Z.N.; data curation, Z.N.; writing—original draft preparation, Z.N.; writing—review and editing, Z.N. All authors have read and agreed to the published version of the manuscript.

**Funding:** This research received no external funding.

**Institutional Review Board Statement:** Not applicable.

**Informed Consent Statement:** Not applicable.

**Data Availability Statement:** Not applicable.

**Conflicts of Interest:** The authors declare no conflict of interest.

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
