# Peer review of "Experimental and Simulation Study on the Effect of Reduction Pretreatment on the Void Healing of Heavy Plate"

_metals, doi:10.3390/met12030400_

Round 1

Reviewer 1 Report

This paper gives the godd information for void closure in sheet rolling process.

  1. In eq.(1), σ is the stress [MPa],  
  2. The values x and y axis are dimensionless In Fig. 9 and 11. What are reference values?
  3. In figue title of Fig. 10, (a)time=0; (b) time=100; (c) time=150; (d) time=175
  4. 4. In Fig. 13, what value is the standard impact energy of 42CrMo steel?

Author Response

We would like to thank the reviewers for their comments, which were very helpful in improving our paper, and we will respond to each of the comments below.

Q1: In eq.(1), σ is the stress [MPa],  

A1: Revisions have been made in the manuscript.

Q2: The values x and y axis are dimensionless In Fig. 9 and 11. What are reference values?

A2: The reference value of the y-axis is w0, which is equal to the actual length, y’, ensuring the dimensionless y=y/W0=1. The reference value for the x-axis is t=W02/Dv, where Dv is the diffusion coefficient of the vacancy in the matrix, but due to the lack of actual diffusion coefficients, the dimensionless parameters from Ref.29 were used directly for the qualitative simulation.

Ref.29: Millett P C, El-Azab A, Rokkam S, et al. Phase-field simulation of irradiated metals: Part I: Void kinetics. Comp Mater Sci, 2011, 50, 949-959.

Q3: In figue title of Fig. 10, (a)time=0; (b) time=100; (c) time=150; (d) time=175

A3: Revisions have been made in the manuscript.

Q4: In Fig. 13, what value is the standard impact energy of 42CrMo steel?

A4: The standard impact energy of 42CrMo is greater than 63 J. Since the purpose of the impact test in this study is mainly to demonstrate the effect of void on impact performance, and there is no process optimization for the impact energy of 42CrMo, the impact energy of the final steel plate is less than the standard impact work of 42CrMo.

Reviewer 2 Report

The presentation and the subject are interesting. The dimensionless representation even appears to be an advantage for a more general statement. In this context, the size relation of the original void size to the grain size resp. in relation to the RVE size could be indicated (Fig. 8 and 9). Please also briefly explain why the void in Fig. 10 shrinks only in its longitudinal axis, but this cannot be observed simultaneously in the transverse axis.

Author Response

We would like to thank the reviewers for their comments, which were very helpful in improving our paper, and we will respond to each of the comments below.

Q: In this context, the size relation of the original void size to the grain size resp. in relation to the RVE size could be indicated (Fig. 8 and 9).

A: The radius of the original void is 10 Δx, the size of the REV is 256 Δx × 256 Δx, Δx=1 is the mesh size of finite element difference method for solving the phase-field model. The grain size varies with the number of grains. Revisions have been made in the manuscript.

Q: Please also briefly explain why the void in Fig. 10 shrinks only in its longitudinal axis, but this cannot be observed simultaneously in the transverse axis.

A: From Equation 2, we can see that the driving force for the void shrinkage comes from two parts, the difference in chemical energy, and the other is the reduction of the interfacial energy due to the curvature difference interface. As can be seen from Figure 10, the interface in the long axis direction has a significant curvature difference. Therefore, it can shrink very quickly, while the interface in the short axis direction is flat with very little curvature difference and does not shrink simultaneously as the long axis. When the length of the long and short axes are about the same, they will start to shrink simultaneously.